# 3D Modeling of Silver Doped ZrO_2_ Coupled Graphene-Based Mesoporous Silica Quaternary Nanocomposite for a Nonenzymatic Glucose Sensing Effects

**DOI:** 10.3390/nano12020193

**Published:** 2022-01-07

**Authors:** Kamrun Nahar Fatema, Chang-Sung Lim, Yin Liu, Kwang-Youn Cho, Chong-Hun Jung, Won-Chun Oh

**Affiliations:** 1Department of Advanced Materials Science & Engineering, Hanseo University, Seosan-si 356-706, Korea; kamrunnahar270@gmail.com (K.N.F.); cslim@hanseo.ac.kr (C.-S.L.); 2Anhui International Joint Research Center for Nano Carbon-Based Materials and Environmental Health, College of Materials Science and Engineering, Anhui University of Science & Technology, Huainan 232001, China; yinliu@aust.edu.cn; 3Korea Institutes of Ceramic Engineering and Technology, Soho-ro, Jinju-si 52851, Korea; kycho@kicet.re.kr; 4Decommissioning Technology Research Division, Korea Atomic Energy Research Institute, Yuseong, Daejeon 305-600, Korea; nchjung@kaeri.re.kr

**Keywords:** glucose sensor, electrocatalytic performance, functional stability, interfering agents, urine

## Abstract

We described the novel nanocomposite of silver doped ZrO_2_ combined graphene-based mesoporous silica (ZrO_2_-Ag-G-SiO_2_,) in bases of low-cost and self-assembly strategy. Synthesized ZrO_2_-Ag-G-SiO_2_ were characterized through X-ray diffraction (XRD), scanning electron microscopy (SEM), energy-dispersive X-ray spectrometry (EDX), transmission electron microscopy (TEM), high-resolution transmission electron microscopy (HRTEM), Raman spectroscopy, Nitrogen adsorption-desorption isotherms, X-ray photoelectron spectroscopy (XPS), and Diffuse Reflectance Spectroscopy (DRS). The ZrO_2_-Ag-G-SiO_2_ as an enzyme-free glucose sensor active material toward coordinate electro-oxidation of glucose was considered through cyclic voltammetry in significant electrolytes, such as phosphate buffer (PBS) at pH 7.4 and commercial urine. Utilizing ZrO_2_-Ag-G-SiO_2_, glucose detecting may well be finished with effective electrocatalytic performance toward organically important concentrations with the current reaction of 9.0 × 10^−3^ mAcm^−2^ and 0.05 mmol/L at the lowest potential of +0.2 V, thus fulfilling the elemental prerequisites for glucose detecting within the urine. Likewise, the ZrO_2_-Ag-G-SiO_2_ electrode can be worked for glucose detecting within the interferometer substances (e.g., ascorbic corrosive, lactose, fructose, and starch) in urine at proper pH conditions. Our results highlight the potential usages for qualitative and quantitative electrochemical investigation of glucose through the ZrO_2_-Ag-G-SiO_2_ sensor for glucose detecting within the urine concentration.

## 1. Introduction

Biosensors are developed for giving symptomatic data for the patient’s prosperity status. Electrochemistry, fluorescence, colorimetry, photoelectrochemistry, and chemical luminescence, have been received for glucose sensing [1,2]. Among them, the electrochemical detecting method has gotten high attention due to its high affectability, promising reaction time [3,4,5]. Glucose oxidation on the sensor is dependable for the chemisorption of the hydroxyl group onto the metal oxide and shaping the bond among the d-electron of metal and glucose atoms. The oxidation state of glucose particles is affected by the metal surface as well as metal-glucose interaction, glucose-metal bond quality, and desorption of glucose particles. By considering an imitating method of the enzyme-like component, a few metals and metal oxides like Au, Pt, Cu, Ni, Mn, Co, and Fe [6,7,8,9,10,11,12,13,14,15,16,17,18,19,20,21] have been studied. Limitation of detection (LOD) for the analyte, the typical nanomaterials considered [22]. Graphene has gotten around the world consideration for the improvement of biosensors because graphene-based biosensors have high electron transfer rates, high charge-carrier mobility, and are extremely significant for biomarkers owing to their extraordinary electrochemical (amperometric, voltammetry, impedimetric) response [23,24]. In addition, graphene shows a thickness of the edge-plane-like structure, giving numerous dynamic destinations for electron transfer to chemical and biological species [23]. Graphene containing zirconium oxide (ZrO_2_) offers a way to upgrade their application by allowing flexible and ideal electrochemical properties, extraordinary potential applications within the broad fields of sensing [25,26,27,28,29,30]. The affectability and conductivity of graphene may be advance upgraded by enhancing Ag NPs owing to their high electron transfer for modifiers in biosensors [31,32]. Biocompatibility, nontoxicity, high conductivity, chemical and steadiness of SiO_2_ make to idealize for utilization for adsorption, biosensors [33,34]. With these points, we developed the ZrO_2_-Ag-G-SiO_2_ which was effectively synthesized by the self-assembly method. ZrO_2_, G, and SiO_2_ have octahedral coordination, Ag occupies on the ZrO_2_-G-SiO_2_ displays giving dynamic response for possible charge transfer to electrolyte [35,36,37].

In this consideration, ZrO_2_-Ag-G-SiO_2_ nanocomposite was developed main active material for glucose sensing. ZrO_2_-Ag-G-SiO_2_ was effectively synthesized by utilizing a basic, low-cost, self-assembly method, and was inspected for nonenzymatic glucose oxidation for the quick response. It shows especially high effectiveness for glucose oxidation counting a greatly low working potential of as it were 0.2 V vs Ag/AgCl. In general, ZrO_2_-Ag-G-SiO_2_ affirmed a significant response without any electron facilitator, provoking a novel way for glucose sensing within the urine. The electrochemical sensing behavior of the ZrO_2_-Ag-G-SiO_2_ sensor towards glucose sensing was examined utilizing amperometric techniques.

## 2. Experimental

### 2.1. Materials

All chemicals used analytical grade without further modification. Graphite powder (99%), zirconium (IV) isopropoxide (70 wt% in 1-Propanol), Pluronic F127 were purchased from Sigma Aldrich (Seoul, Korea). Ethylene Glycol, AgNO_3_, HCl, Phosphate Buffer, NaOH, KOH, Ethylene Glycol were purchased from Dae-Jung Chemical Korea (Busan, Korea). Deionized water (18.2 MΩcm^−1^) was a self-made product.

### 2.2. Synthesis of ZrO_2_

6.5 g of Pluronic F127 was mixed up in 50 mL of ethanol and zirconium (IV) isopropoxide solution was included in 50 mL of ethanol and ethylene glycol separately with vigorous mixing and added together at 314 K with 50 mL of H_2_O. Hydrochloric acid was included to alter the pH 2.4 and kept at 314 K for 1 h and 354 K in a closed container for 24 h after that dried at 374 K and calcined at 674 K for 5 h.

### 2.3. Synthesis of Silver Doped ZrO_2_ (ZrO_2_-Ag)

3.5 g of AgNO_3_ was in 50 mL of deionized water. Then ZrO_2_ was poured dropwise to solution blended till the gel came out. The gel was dried at 374 K for 3·1⁄2 h, calcined at 674 K, and after that ground to get the ZrO_2_-Ag nanoparticles.

### 2.4. Synthesis of ZrO_2_-Ag-G

0.33 g of graphite oxide (GO) was scattered into 300 mL of water and ultra-sonicated for 40 min. Sonicated graphene oxide exchanged poured into ZrO_2_-Ag solution and 50 mL of 1 M sodium hydroxide included into the sonicated mixture dropwise for expected pH and blended for 3 h at 374 K. The color turned into coffee color, demonstrates the effective combination of G with Ag combined ZrO_2_ arrangement E.

### 2.5. Synthesis of ZrO_2_-Ag-G-SiO_2_

For the synthesis of final nanocomposites, 1.1 g of triblock copolymer Pluronic F-127 was included to 100 mL of deionized water and 61 mL of 2 M HCl at 313 K. 4 mL of tetraethyl orthosilicate (TEOS) was included and blended at 314 K for 12·1⁄2 h and heated to 374 K for 20 h after that washed with water and ethanol and dried at 338 K overnight and the copolymer was calcination at 824 K for 3·1⁄2 h. The solution ZrO_2_-Ag-G was drop-by-drop included on 0.3 g of the silica powder and this mixture was blended with 374 K for 24 h and ultrasonicated for 1_1⁄2_ h and get the powder, washed with 1.5 mL of methanol, and dried at 338 K overnight. Calcined at 974 K at 283 K/min and held at 974 K for 5 h. Dark color items were found. 

### 2.6. Preparation of ZrO_2_-Ag-G-SiO_2_ Electrode

The zrO2-Ag-G-SiO2 coated film was prepared using a routine doctor-blade method [38]. For the altered doctor-blade method, we controlled the thickness of ZrO_2_-Ag, ZrO_2_-Ag-GO, ZrO_2_-Ag-GO-SiO_2_. To begin with, synthesized fabric powder (1.1 g) was mixed with Ethylcellulose and acetone (1.5 mL) in a mortar for 15 min. After that, the prepared glues were coated on FTO glass to create a film, after being dried within the open state for 35 min. One drop greasing up oil was put onto the film surface and stabilized beneath 374 K in the dry oven for 25 min to decrease breaks.

### 2.7. Characterization of the Materials

The phase structure and purity of as-synthesized products were examined by X-ray diffraction (XRD, Rigaku, Chiba, Japan) with Cu-Kα radiation (λ = 1.5406 Å) at 40 kV, 30 mA over 2θ range of 20–70. Morphologies were studied utilizing scanning electron microscopy (SEM) and EDS analysis by utilizing an SEM (JSM-76710F, JEOL, Tokyo, Japan), a transmission electron microscopy (TEM) (JEM-4010, JEOL, Tokyo, Japan), and a high-resolution TEM (HRTEM) (JSM-76710F, JEOL, Tokyo, Japan) operated at 300 kV accelerating voltage. X-ray photoelectron spectroscopy (XPS), Diffuse Reflectance Spectroscopy (DRS, SolidSpec-3700, Tokyo, Japan), and Raman spectroscopy (RAMAN, LabRAM HR-800, Chiba, Japan) analyses were performed by utilizing WI Tec. alpha 300 series. Porous characterization of ZrO_2_-Ag-G and ZrO_2_-Ag-G-SiO_2_ structures was performed by a full analysis of N_2_ adsorption/desorption tests (BELSORP-max, BEL Japan Inc., Tokyo, Japan). (PG201, Potentiostat, Galvanostat, Volta lab ^TM^, Radiometer, Aalborg, Denmark).

### 2.8. Electrochemical Measurements

Cyclic voltammetry (CV) and estimations were performed a three-electrode electrochemical set up to check the current and voltage profiles, where ZrO_2_-Ag, ZrO_2_-Ag-G, ZrO_2_-Ag-G-SiO_2_ was utilized as working electrode whereas platinum and Ag/AgCl electrode as counter and reference anodes, individually. Electrochemical properties in commercial urine were utilized with the measured pH 6.0, 6.7, and 6.5, individually. As electrolytes, 0.1 M NaOH, 0.1 M KOH, and Buffer were utilized. The following equation is used to determine the *LOD* [39,40,41]
*LOD* = 3 *SD*/*N*(1)
where *SD* is the standard deviation of the analyte concentration calculated from the current reaction of progressive including of glucose into the electrolyte; *N* is the slope of the calibration curve which demonstrates the affectability of the anode with a signal-to-noise ratio 3. Moreover, CV tests were performed from −0.3 to +0.2 V versus Ag/AgCl at a filter rate of 10 mV s^−1^. All estimations were carried out by voltammetry (PG201, Potentiostat, Galvanostat, Volta lab ^TM^, Radiometer, Aalborg, Denmark).

## 3. Results

### 3.1. Characterization of the ZrO_2_-Ag-G-SiO_2_ Sample

The mesoporous semiconductors were anchored on graphene nanosheets since this mesoporous conductive arrangement facilitates electron transport among nanostructure and electrolytes, hence making this a desirable stage for the design of biosensors. Figure 1 illustrates the crystalline characteristic properties of ZrO_2_-Ag, ZrO_2_-Ag-G, and ZrO_2_-Ag-G-SiO_2_ samples affirmed by the X-ray diffraction (XRD) technique.

The XRD patterns of the nanoparticles give the major 2y peak values at 25.6, 30.2, 32.9, 38.5, 44.6, 46.2, 50.2, 54.5, 55.2, 63.8, 76.4, and 84.5 in accordance with monoclinic zirconia. All diffraction peak values matched with the international standard file (JCPDS 37-1484). After modification, Ag nanoparticles could be indexed based on pure silver oxide having the symmetry of face center cubic. The peaks designated to the planes with the hkl values of 38.5 (111), 63.8 (131), 74.39 (220), and 80.58 (311), respectively, are the same with the XRD pattern of the (JCPDS 65-2871). After modification with SiO_2_ the ZrO_2_-Ag-G-SiO_2_ all diffraction peaks along the (JCPDS 39-1425), affirming the crystalline nature of the samples.

The particle composition was also studied by TEM and SEM and the images are shown in Figure 2a–d.

Figure 2a showed the morphology of profoundly amplified TEM image of ZrO_2_ which distributed as clustered in a flower shape. Morphology of ZrO_2_ was also confirmed by SEM image (Inset). Figure 2b showed the TEM image of ZrO_2_-Ag where Ag nanoparticles interconnected with ZrO_2_. Figure 2c revealed the good distribution of ZrO_2_-Ag on the Graphene surface. Figure 2d showed that the ZrO2-Ag G combined with mesoporous SiO_2_. These figures showed that ZrO_2_-Ag-G-SiO_2_ were uniformly distributed. Every single TEM image is carried with a corresponding SEM image (Inset). Such flake-like nanostructured geometry leads to a rough surface of the electrode which can expectedly lead to an upgrade of the electrode performance on account of its high surface area, high surface-to-volume ratio, and exposure of more active sites on ZrO_2_-Ag-G-SiO_2_. Figure 2e,f showed the HRTEM image, the lattice space of 0.28 nm was given out to the interplanar of the (111) plane of the ZrO_2_-Ag-G-SiO_2_ sample and another lattice space of 0.26 nm was arranged to the interplanar of the (022) plane of the ZrO_2_-Ag-G-SiO_2_ sample.

The elemental state of the ZrO_2_-Ag-G-SiO_2_ nanoparticles was furthermore analyzed through EDS mapping.

As shown in Figure 3, the composition of ZrO_2_-Ag-G-SiO_2_ was presented to confirm the coexistence of Zr, C, Ag, and Si with the evaluated composition within the gravimetric rate of 29% Zr, 35% C, 12% Ag, and 5% Si.

Raman spectroscopy was also performed to characterize the G band showing in the composite.

As shown in Figure 4a, the G band of the as-synthesized sample appeared two peaks located at 1331 and 1573 cm^−^^1^ corresponding to the (D band) and the C-C bond stretching frequency (G band), individually. For the most part, the intensity ratio of the D- and G bands (ID/IG) is utilized to evaluate the degree of disorder and the average size of sp^2^ spaces. In this fact, the value of ID/IG was calculated to be 0.94.

The resultant absorbance of UV-DRS is depicted in Figure 4b. The optical bandgap of the ZrO_2_-Ag, ZrO_2_-Ag-G, ZrO_2_-Ag-G-SiO_2_, can be determined by the (Equation (2)): [42]
(2)(αhv)12=A(hv−Eg)
where ‘*α*’ was the molar assimilation coefficient calculated as *α* = (1 − R)^2^/2R, *hʋ* is the incident light frequency, ‘*A*’ is the proportionality constant, and ‘*E_g_*’ is the bandgap energy of the material. Table 1 outlines the information form (*αhv*)^1⁄2^ as a function of photon energy. Band gaps showed 3.11 eV for Ag-doped ZrO_2_-Ag and decrease after combining with graphene turned to 2.61 for the ZrO_2_-Ag-G eV. Surprisingly, the band gaps change remarkably decreased to 2.00 eV within the ZrO_2_-Ag-G-SiO_2_ after combining through mesoporous SiO_2_. Valence band (VB) and conduction band (CB) potentials of all the samples were calculated based on the following equations [43]
(3)$$E_{\mathrm{CB}}\;=X-\mathrm{Ee}-\frac{1}{2}\;E_{g}\;$$
 E_VB_ = E_CB_ + E_g_(4)


Here, E_VB_ and E_CB_ are valence and conduction band edge potentials, individually. c is the electronegativity of the semiconductor, E_e_ is the energy of free electrons on the hydrogen scale and E_g_ is the bandgap energy of the semiconductor.

Figure 4c presents the N_2_ adsorption-desorption isotherms of ZrO_2_-Ag-G and ZrO_2_-Ag-G-SiO_2_ samples. ZrO_2_-Ag-G and ZrO_2_-Ag-G-SiO_2_ samples display typical type IV isotherm, illustrating those materials had mesopores. Isotherms of samples display an H_2_ type hysteresis loop at a relative pressure (P/P_0_) between 0.6 and 0.9, showing that these materials possess large and uniformly distributed mesopores. In addition, the hysteresis loops gradually shift to higher relative pressure (P/P_0_) from ZrO_2_-Ag-G to ZrO_2_-Ag-G-SiO_2_, proposing that these mesopores were extending with the counting mesoporous SiO_2_. A mesopore diameter as large as 5.67 nm finds out the ZrO_2_-Ag-G sample. When combining through mesoporous SiO_2_ it proceeds to extend up to 8.96 nm as well as BET surface area also expanded from 8.66 to 9.17 m^2^ g^−1^, individually (Table 1). The electrochemical properties of nanocomposites were correlated with the BJH and BET analysis results. From BET analysis, the total pore volume and mean pore diameter of sensor active material are reduced due to the oxidizing agent treatment. According to the summary results of BET and BJH, the surface area and total pore mass of the graphene increased with SiO_2_. The mesopore state and high surface area are the main parameters that are valuable for framing ion-transport tunnels in electrochemical reactions.

In magnetic field determination, one-level effective mass approximation (EMA) is utilized for a basic non-degenerate energy band. Bloch electrons in an energy band are treated as free electrons with the free electron mass m0 replaced by the effective mass m*. The Schrodinger equation for the function of the conduction electron in electric and magnetic fields can be shown with the following equation [43].
(5)$$B=\frac{\mu 0I}{2\pi r}$$

Figure 4d confirms the magnetic field curve of the ZrO_2_-Ag-G-SiO_2_ samples measured at ambient temperature. The saturation magnetization (MS), which is determined by the plot of M versus 1/H using data at low magnetic fields, is observed to be 0.0036.5 emu g^−^^1^ to 0.0046.5 emu g^−^^1^.

For characterizing detailed surface chemical compositions of ZrO_2_-Ag-G-SiO_2_, XPS analysis was performed.

The results are revealed in Figure 5. The complete spectrum of ZrO_2_-Ag-G-SiO_2_ shows the presence of Si, Zr, C, Ag, and O atoms attributed to the effective modification. The corresponding high-resolution spectra with respect to C1s signal 284.5 eV as a reference binding energy in Appendix A attributed to C-C, bonds of graphene. As existing in Appendix A, Si2P peaks were found at 102.8 eV. These peaks located at 184.08 eV correspond to Zr3d in Appendix A. Besides, the interaction of the carbonyl group and hydroxyl group were also confirmed in O1s existing in Appendix A with the binding energy at 531.6 eV corresponding to C-O bonds. Finally, the peaks at 367.0 eV and 373.1 eV revealed in Appendix A correspond to Ag 3d. The overall results of the XPS study confirmed that all surface chemical compositions of ZrO_2_-Ag-G-SiO_2_ were found in the as-prepared nanocomposite.

### 3.2. Electrocatalytic Activity of the ZrO_2_-Ag-G-SiO_2_ Electrode towards Glucose Sensing

The electrochemical tests for working electrodes, ZrO_2_-Ag, ZrO_2_-Ag-G, ZrO_2_-Ag-G-SiO_2_ were performed in a three-electrode cell system with Pt wire as counter electrode and Ag/AgCl as a reference electrode within the potential range of −0.3 to +0.3 V. Figure 6a presents the CV profile of electrochemical responses in 10.5 mL of commercial urine and different electrolytes without glucose.

There was a poor oxidation peak observed in Figure 6a in the absence of glucose. In contrast, ZrO_2_-Ag, ZrO_2_-Ag-G, ZrO_2_-Ag-G-SiO_2_ electrodes showed a well-defined oxidation peak at the potential of +0.2 V. By adding 0.05 mmol/L of glucose, a poor response was noticed with the ZrO_2_-Ag rather than ZrO_2_-Ag-G and ZrO_2_-Ag-G-SiO_2_ electrode in the presence of glucose, due to the high bandgap energy of ZrO_2_. After combining with Ag nanoparticle and graphene, the bandgap energy-reduced and ZrO_2_-a Appendix A for ZrO_2_, rapidly transporting electrons during the electrochemical reaction due to their good conductive property. ZrO_2_-Ag-G-SiO_2_ electrode exhibited a substantial increase in anodic current density 4.0 × 10^−^^3^ mAcm^−^^2^ as showed in Figure 6b. For varying electrolytes such as 0.1 M phosphate buffer, NaOH, KOH, significant and fast current responses 9.0 × 10^−^^3^ mAcm^−^^2^ were observed for the ZrO_2_-Ag-G-SiO_2_ electrode with the addition of 0.55 mmol/L glucose as presented in Figure 6c. The obtained result clearly recommends the oxidation peak corresponds to the electro-oxidation of glucose at the ZrO_2_-Ag-G-SiO_2_ electrode [44]. Thus, a mechanism of non-enzymatic glucose sensing on the ZrO_2_-Ag-G-SiO_2_ electrode is clarified in Figure 1.

To demonstrate the analytical parameters (for example sensitivity, linear range, detection limit, and response time), the amperometric response of the ZrO_2_-Ag-G-SiO_2_ electrode was performed at a fixed voltage of +0.2 V (versus Ag/AgCl) in 0.1 M PBS by stepwise adding of glucose at different concentration. A well-defined and fast response to the ZrO_2_-Ag-G-SiO_2_ electrode was observed. Figure 6d confirms the current response which was estimated to be as high as 5.0 × 10^−^^3^ mA cm^−^^2^ at lower glucose concentration (0.05 mmol/L to 0.35 mmol/L). By adding glucose, the current response quickly reached a steady-state and attains ~98% of response within 1 s. The response current was linearly increased with increasing glucose concentration, ZrO_2_-Ag-G-SiO_2_ electrode exhibited high sensitivity in the linear range (0.05 mmol/L to 0.35 mmol/L).

Appendix A displays the ZrO_2_-Ag-G-SiO_2_ glucose sensor calibration curve and cyclic voltammogram. With a linear range of 150–350 L and a correlation coefficient (R) of 0.996, the calibration curve indicated excellent linearity. Appendix A compares the detecting characteristics of several electrochemical glucose sensors, differentiating LOD and linear range. For glucose oxidation, the ZrO_2_-Ag-G-SiO_2_ sensor has a better linear response range and detection limit. The nanostructure of ZrO_2_-Ag-G-SiO_2_ offered greater surface area and exposed active sites, which facilitated electrolyte transport from solution to all active sites [45,46,47,48,49].

Overall, the enhanced sensing performance of the non-enzymatic glucose sensor is ascribed to the direct growth of mesoporous ZrO_2_-Ag-G-SiO_2_ thin film on FTO electrodes which offers a high surface area for ZrO_2_ modification, resulting in fast electron transfer during the electrochemical process of glucose oxidation occurring between electrolyte and electrode. Importantly, we have used the self-assembly method to fabricate non-enzymatic ZrO_2_-Ag-G-SiO_2_ glucose-sensing electrodes which account for controllable nanostructures with great reproducibility and a cost-effective fabrication process for stable glucose sensing devices.

### 3.3. Selection of Electrolytes towards ZrO_2_-Ag-G-SiO_2_ Electrode

Sensing of glucose by ZrO_2_-Ag-G-SiO_2_ sample with different electrolytes (PBS, NaOH; KOH) and different concentrations was investigated under ambient conditions. Glucose oxidation with these electrolytes was measured in 0.1 M NaOH, phosphate buffer, and KOH by subsequent addition of 0.55 mmol/L of glucose at regular intervals and observed the current responses after every injection.

Figure 7a shows that when 0.55 mmol/L of glucose adding to different concentrations of electrolytes resulted in almost the best current density towards the phosphate buffer electrolytes. The current state of the ZrO_2_-Ag-G-SiO_2_ electrode greatly depends on glucose concentration and electrolyte pH (i.e., the amount of OH^−^), since OH^–^ are required to neutralize the protons generated during the dehydrogenation stage of the reaction. Hence, a better outcome is confirmed towards phosphate buffer for the ZrO_2_-Ag-G-SiO_2_ electrode.

As described above, the sensitivity and linear range of glucose sensing can be found by plotting peak current density against glucose concentrations as shown in Figure 7d. In 0.05 mmol/L glucose concentration, the sensor response had a sensitivity of 4.0 × 10^−^^3^ mA cm^−^^2^ and 0.35 mmol/L glucose concentration, the sensor response had a sensitivity of 5.0 × 10^−^^3^ mAcm^−^^2^. Here we can see that the response range is proportional to the concentration range. So, after observing this ratio we can easily reach this decision that we can measure a diabetic urine sample with this sensor for qualitative and quantitative analysis.

### 3.4. Anti-Interference Ability of the ZrO_2_-Ag-G-SiO_2_ Sensor

The anti-interference ability of non-enzymatic-based glucose sensing devices is a major challenge, which could affect the electrode’s sensing performance. To check the selectivity of ZrO_2_-Ag-G-SiO_2_ electrode in the presence of interfering species (such as Vitamin C, Starch, Lactose, Fructose, NaCl, KCl, and Urea), the amperometric response of the sensing electrode was checked by adding 0.91 mmol/L glucose and each above mentioned interfering species was in same concentration in the 0.1 M PBS solution at +0.2 V (versus Ag/AgCl), shown in Figure 7b. The addition of 0.91 mmol/L glucose leads to a rapid current response, although interfering species addition exhibited negligible current responses. As shown in the histogram of each interfering species addition and current response is shown here, which confirms the negligible current responses compared to 0.91 mmol/L glucose. These results suggest the suitability of the ZrO_2_-Ag-G-SiO_2_ electrode for the selective sensing of glucose in real samples. This confirmed that the ZrO_2_-Ag-G-SiO_2_ electrode was selective towards glucose without being affected by interferences. This enhanced sensing performance is basically attributed to a great interaction among the nanostructure and electrode with the high surface area for catalytic sites, facilitating a suitable path for electron transport during electrochemical activity. The results obtained with the proposed method were compared with other methods for the detection of glucose (Appendix A). Overall, the ZrO_2_-Ag-G-SiO_2_ electrodes can be envisioned as a promising design for non-enzymatic glucose measurement in real clinical samples which may gain considerable benefits for different biomolecules sensing.

## 4. Discussion

The electrocatalytic properties of ZrO_2_-Ag-G-SiO_2_ were examined toward applications involving physiological pH, such as the detection of Glucose. Considering that glucose can be oxidized to gluconolactone (Figure 1) at a neutral pH via a two-electron electrochemical reaction [50,51,52]. However, an excellent response was observed with the ZrO_2_-Ag-G-SiO_2_ sensor in the presence of glucose. This can be attributed to the excellent electrocatalytic nature of ZrO_2_, which mediates the heterogeneous chemical oxidation or reduction of the glucose, while the converted ZrO_2_ can be continuously and simultaneously recovered by electrochemical oxidation or reduction due to their high surface to volume ratio [51]. Additionally, in our sensor, the Ag-G-SiO_2_ works as a Appendix A for ZrO_2_, rapidly transporting electrons during the electrochemical reaction due to their good conductive property. Also, the less dense morphology of the Ag-G-SiO_2_ provides better permeability of the sensing matrix to the solution. The possible electrochemical reactions involved in glucose oxidation through the Zr^4+^/Zr^3+^ centers of ZrO_2_ are given below [51]: Zr^4+^ + Glucose (C_6_H_12_O_6_) → Zr^3+^ + Gluconolactone (C_6_H_10_O_6_) + H_2_O (6)
Gluconolactone (C_6_H_10_O_6_) + H_2_O → 2H^+^ + Gluconate (C_6_H_12_O_7_)(7)
2Zr^3+^ → 2Zr^4+^ + 2e^−^(8)

Therefore, electrooxidation of glucose on ZrO_2_-Ag-G-SiO_2_ for the nonenzymatic detection of glucose at physiological pH was investigated.

## 5. Conclusions

We developed a simple approach for producing ZrO_2_-Ag-G-SiO_2_ using the facile self-assembly method, producing a catalyst-coated and binder-free composite electrode. The ZrO_2_-Ag-G-SiO_2_ exhibited a uniform and highly mesoporous network of the catalytic film. Also, multiple active sites in ZrO_2_-Ag-G-SiO_2_ along with enhanced conductivity of graphene oxide improved the electrocatalytic performance of this electrode toward glucose oxidation. The ultra-high sensitivity (9.0 × 10^−^^3^ mA cm^−^^2^) at a low applied potential of only 0.2 V versus Ag/AgCl, wide linear range (0.05 mmol/L–0.35 mmol/L), low sensing limit (0.05 mmol/L), with impressive qualitative and quantitative analysis, selectivity and stability make this ZrO_2_-Ag-G-SiO_2_ a promising electrode to serve as a non-enzymatic glucose sensor. Based on the results, ZrO_2_-Ag-G-SiO_2_ provided an excellent sensitivity in commercial urine specimens, so this biosensor is believed to have a high possibility for practical use.

## Data Availability

Data are contained within the article.

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
