# Peer review of "3D Modeling of Silver Doped ZrO2 Coupled Graphene-Based Mesoporous Silica Quaternary Nanocomposite for a Nonenzymatic Glucose Sensing Effects"

_nanomaterials, 2022, doi:10.3390/nano12020193_

Round 1

Reviewer 1 Report

All the Reviewer's comment are successfully addressed. Authors made various changes in revised resubmitted manuscript. It can be accepted for publication in present form. 

Author Response

Thank you so much..

Reviewer 2 Report

The authors have considerably improved the writing of the manuscript and now it is relatively easy to follow. As for the characterization of the electrode modification, it is quite extensive and rigorous. However, in my opinion, the application to the determination of glucose still needs more work, as for instance, a calibration graph and the calculation of analytical parameters such as the detection limit, the sensitivity or the linearity range. Also, these parameters should be compared in a table with previous methods in the literature (quite abundant in the case of glucose detection). Hence, a major revision about these points is required before the manuscript can be accepted for publication.

Author Response

The authors have considerably improved the writing of the manuscript and now it is relatively easy to follow.

As for the characterization of the electrode modification, it is quite extensive and rigorous.

However, in my opinion, the application to the determination of glucose still needs more work, as for instance, a calibration graph and the calculation of analytical parameters such as the detection limit, the sensitivity, or the linearity range.

Also, these parameters should be compared in a table with previous methods in the literature (quite abundant in the case of glucose detection). Hence, a major revision about these points is required before the manuscript can be accepted for publication.

: We did the calibration graph according to your comments as follow:

 Figure S4 displays the ZrO2-Ag-G-SiO2 glucose sensor calibration curve and cyclic voltammogram. With a linear range of 150-350 L and a correlation coefficient (R) of 0.996, the calibration curve indicated excellent linearity. Table S1 compares the detecting characteristics of several electrochemical glucose sensors, differentiating LOD and linear range. For glucose oxidation, the ZrO2-Ag-G-SiO2 sensor has a better linear response range and detection limit. The nanostructure of ZrO2-Ag-G-SiO2 offered greater surface area and exposed active sites, which facilitated electrolyte transport from solution to all active sites.[52]

 S4. Calibration curve of the ZrO2-Ag-G-SiO2 glucose sensor.

Table S1. Comparison of glucose detection method using various electrochemical sensing electrodes

Detection method

Material

LOD

Linear range

(mmol/L)

Reference

Electrochemical

Ni-SnOx/PANI/CuO

0.130 mmol/L

1-10 mmol/L

[49]

Fluorimetry

Mn-doped Zn0.5Cd0.5@ZnS NRs

0.1 mmol/L

0.05-0.3 mmol/L

[50]

Colorimetry

Cu2(OH)3Cl-CeO2 NPs-TMB

0.05 mmol/L

0.1-2 mmol/L

[51]

Electrochemical

Nafion/GOx/ZnO/FeC11SH/Au

-

0.05-1.0 mmol/L

[52]

Microfluidic

Gel-encapsulated B5

0.9 mmol/L

1.0–50 mmol/L

[53]

Electrochemical

ZrO2-Ag-G-SiO2

22.13 µL

150-350 µL

This work

(I) Any revisions to the manuscript should be marked up using the “Track 
Changes” function if you are using MS Word/LaTeX, such that any changes can 
be easily viewed by the editors and reviewers. 
(II) Please provide a cover letter to explain, point by point, the details of 
the revisions to the manuscript and your responses to the referees’ 

comments. 
(III) If you found it impossible to address certain comments in the review 
reports, please include an explanation in your rebuttal. 
(IV) The revised version will be sent to the editors and reviewers. 

Round 2

Reviewer 2 Report

The authors have revised the manuscript according to my suggestions and the results are fine.

This manuscript is a resubmission of an earlier submission. The following is a list of the peer review reports and author responses from that submission.

Round 1

Reviewer 1 Report

3D Modeling of Silver Doped ZrO2 Coupled Graphene-based 2 Mesoporous Silica Quaternary Nanocomposite for a Nonenzy-3 matic Glucose Sensing Effects

 After thorough editing of the manuscript, a review was possible. English still needs thorough revision, the manuscript cannot be published as it is.

1-     The introduction is very difficult to understand due to the language. It seems that the text was produced from an automatic translation which causes many wrong translations.

2-     The same for Materials and Methods

3-     Figure 1 is difficult to read because the XRD patterns and the (hkl) values overlap

4-     Figure 1 is not sufficiently commented on in the text.

5-     Figure 2 caption must be completed with all details

6-      Results of Figure 2 are not commented on. What are the conclusions from the analysis?

7-     Discussion from Figure 5 and Table 1 cannot be understood as it is.

8-     Figure 6 (XPS) is not presented as it should be. Please refer to any other published article in the literature where XPS results are presented and discussed.

9-      From Figure 7 (CVs), it appears that the conductivity of the material is exceedingly low, which makes CVs deformed, tilted, mostly showing resistivity rather than electron transfer. These CVs cannot be taken as a basis for discussion on the catalytic oxidation of glucose

10- “PO43- buffer” is not a buffer and does not give a pH of 7.4. The authors must correct this.

11- Figure 7 shows the current as a function of time and potential. It is not the correct way to present a CV. The authors should, if necessary, show chronoamperometry but not as it is made and shown here

12- Figure 8 is not supported by any discussed results. What are the amperometry conditions? No results are provided.

13- Scheme 1 is not indicative and should be removed.

14- The conclusion cannot be understood as it is.

As a conclusion, the manuscript must be thoroughly edited, rewritten by an English speaking writer, and the results rearranged and explained, before a resubmission.

Reviewer 2 Report

The manuscript reports the synthesis and application of a nanocomposite Fe(III) detection. Although providing some interesting results, in my opinion, the manuscript is not suitable in the present form to be published in Nanomaterials.

Some specific comments:

  • The overall English used in the manuscript needs several improvements. Sentence construction is poor with many grammatical errors, which make the text difficult to understand.
  • Abstract – Line 17: What the authors mean with “minimal expense”?
  • Abstract – Line 19: please, check the names of the techniques.
  • Introduction section: The manuscript lacks an adequate use of the recent published literature to introduce the background of this study.
  • Experimental section – Line 141: Please, provide more details about the electrochemical measurements.
  • Section 3.1: The physicochemical characterization appears to be redundant. Some of the characterization data should be omitted or moved to the supplemental section.
  • Section 3.1 – Figure 2:The EDS mapping analysis should be provided to confirm the attributions made by the authors.
  • Section 3.2 – Figure 7: Present only one of the curves for each of the solutions or materials.
  • Section 3.3 – Figure 8: Error bars are missing.

Reviewer 3 Report

Present manuscript reports the synthesis of  silver doped ZrO2- graphene-
mesoporous silica (ZrO2-Ag-G-SiO2, ZAGS) composite for application in glucose sensing via CV. Manuscript is timely and has good point in characterization. It can be accepted for publication after addressing the following concern.

  1. There need a comparative study in investigating the glucose sensor with control ZrO2, graphene, Ag NPs and SiO2.
  2. Selectivity should be tested with similar molecules like fructose, dextrose, sucrose and carbohydrates.
  3. To further support the mechanism trap study or EPR measurements could be performed.
  4. Several closely related references could be cited: ACS Biomaterials Science & Engineering, 2020, 6, 5527–5537; RSC Advances, 2016, 6, 37319-37329. 
  5. A comparative table with  other NPs having glucose sensing ability should be included.
  6. There are some grammar error. The author should carefully proofread the manuscript.

Reviewer 4 Report

The English of the manuscript is poor (it looks like an automatic translation without further revision) and in some parts it is really difficult to understand what the authors are explaining, whith some curious expressions such as 'decent voltage' (line 322) or 'impartial pH' (line 363). Formally it has also many flaws: no subscripts or superscripts, an excessive number of figures too large and divided in some pages... As for the scientific content, there is an exhaustive characterization of the new material by using many techniques, but the study of its application to the detection of glucose should be more accurate and complete (Figure 7d is far from a calibration line and the calculations of the detection limit and the linearity range are unclear). In conclusion, I think that the manuscript in its present form is not acceptable for publication in Nanomaterials.